EGCG’s anticancer potential unveiled: triggering apoptosis in lung cancer cell lines through in vitro investigation

Khair Al Moutassem Billah 1 2
Maniangat Luke Alexander 1 2
Patnaik Rajashree 1
Testarelli Luca luca.testarelli@uniroma1.it 3
1 Department of Basic Medical & Dental Sciences, College of Dentistry, Ajman University , Ajman , United Arab Emirates
2 Centre of Medical and Bio-allied Health Science Research, Ajman University , Ajman , United Arab Emirates
3 Department of Oral and Maxillo-Facial Sciences, Sapienza University of Rome , Rome , Italy
van der Westhuizen Francois
Electronic publication date: 2025 Mar 26
Publication date: 2025
Volume: 13
Electronic Location ID: e19135
Received 2024 May 1; Accepted 2025 Feb 19
Copyright: ©2025 Khair et al.
Copyright year: 2025
Copyright holder: Khair et al.
License: This is an open access article distributed under the terms of the Creative Commons Attribution License, which permits unrestricted use, distribution, reproduction and adaptation in any medium and for any purpose provided that it is properly attributed. For attribution, the original author(s), title, publication source (PeerJ) and either DOI or URL of the article must be cited.
License URL: https://creativecommons.org/licenses/by/4.0/

Keywords: A549 cells, Apoptosis rate, Cell proliferation, Epigallocatechin -3- gallate (EGCG), H1299 cell, Lung cancer, Protein expression

Funding: Ajman University Internal Research 2020-IRG-DEN-04 This research was supported by Ajman University Internal Research Grant No. (DRG Ref. Number- 2020-IRG-DEN-04). The funders had no role in study design, data collection and analysis, decision to publish, or preparation of the manuscript.

==============================
Background

Novel treatment techniques are needed since lung cancer is still a major worldwide health concern. Green tea contains a component called epigallocatechin-3-gallate (EGCG), which has demonstrated potential anticancer properties. This work sought to understand how EGCG affects the phosphatidylinositol-3-kinase protein kinase B (PI3K/Akt) signaling pathway, which in turn causes apoptosis in H1299 lung cancer cells.

Methods

In this experiment, multiple dosages of EGCG were applied to five H1299 cells and five A549 cell lines for a duration of 72 h. Apoptotic pathways, cellular responses, and protein expression levels were investigated in relation to EGCG by morphological, biochemical, and proliferation/migration investigations.

Results

In H1299 and A549 cells, EGCG raised apoptosis rates and, in a dose-dependent way, hindered cell growth. The levels of phosphorylated Akt (p-Akt) and PI3K (p-PI3K) dramatically reduced following EGCG administration, despite no significant alterations in Akt and PI3K expressions. These results imply that EGCG inhibits the activation of the PI3K/Akt signaling pathway, which in turn causes apoptosis in H1299 and A549 cells.

Conclusion

The research provides insights into the effects of EGCG on proliferation and migratory inhibition, as well as highlighting its potential to induce apoptosis in lung cancer cells. These results support EGCG’s promise as a therapeutic agent in the treatment of lung cancer and further our understanding of the processes underlying its anticancer activities.

Introduction

With differing rates among populations and countries, lung cancer is a worldwide health problem. Incidence rates of adenocarcinoma among young women are on par with or higher than those among young males in a number of nations, including the United States (Jemal et al., 2018), Canada, Denmark, Germany, New Zealand, and the Netherlands (Fidler-Benaoudia et al., 2020). The incidence of adenocarcinoma continues to rise, which might result in greater death rates (Cheng et al., 2016).

Apoptosis plays a critical role in the aetiology and management of cancer, in addition to being essential for maintaining cellular homeostasis. Studies highlight apoptosis as a vital tissue balance regulator, with dysregulation resulting in cancer (Chaudhry et al., 2022). Anticancer medicines are responsive to both intrinsic (mitochondrial) and extrinsic (death receptor) pathways, and abnormalities in both pathways lead to treatment resistance.

In order to destroy cancer cells or make them more susceptible to other treatments, novel therapeutics target certain apoptosis-controlling molecules and pathways (Kim, Kin & Beck, 2024). Cancer treatment is significantly hampered by resistance to apoptosis, despite continuous attempts to comprehend its molecular processes and devise countermeasures. Apoptosis is a target for treatment as well as a preventative process in the context of cancer. Research is ongoing to increase the effectiveness of cancer treatments by selectively causing cancer cells to undergo apoptosis (Morana, Wood & Gregory, 2022).

Evidence suggests that the tyrosine kinase inhibitor (TKI) BIBW2992 targets the PI3K/AKT signalling pathway. This drug effectively overcomes T790M-EGFR-mediated erlotinib resistance in H1975 human lung cancer cells by inhibiting the PI3K/AKT signalling pathway. Combination of BIBW2992 and ARQ 197 is effective against erlotinib-resistant human lung cancer cells with the EGFR T790M mutation (Jiao et al., 2018). Further, newer biological agents targeting the NSCLC oncogene have gained approvals recently (Paglialunga, Ricciardi & D’Arcangelo, 2018).

Contemporary approaches to lung cancer treatment encompass a spectrum of interventions, ranging from conventional methods to innovative modalities. Ongoing research investigates multimodal strategies like neoadjuvant chemotherapy, concurrent chemo-radiotherapy, and post-operative irradiation, particularly for stages II and III NSCLC (Chang & Chang, 2022). Notably, the integration of traditional Chinese medicine introduces a holistic dimension to treatment exploration. The optimal course of action varies based on the specific stage of lung cancer, with diverse combinations demonstrating efficacy in terms of response rates and disease-free control rates (Li, Feiyue & Gaofeng, 2021). Hence, personalized and stage-specific treatment strategies are warranted for patients with lung cancer (Antonoff, Antonoff & D’Cunha, 2012).

Green tea contains a powerful polyphenol called epigallocatechin gallate (EGCG), which is well-known for its anticancer effects. Studies have demonstrated that EGCG suppresses several cancer forms, including triple negative breast cancer (TNBC). Research has shown that in cancer cells, EGCG stops the cell cycle, causes apoptosis, and slows cell proliferation (Bimonte et al., 2020). Furthermore, new compounds such as Pro-EGCG have been created to improve the stability and efficacy of EGCG in preventing tumour development and triggering apoptosis in breast cancers (Marín et al., 2023).

Beyond breast cancer, other malignancies such as ovarian, pancreatic, gastric, lymphoma, lung, and myeloma can also benefit from EGCG’s therapeutic value (Kciuk et al., 2023). It has been demonstrated to halt the development and proliferation of cancer cells, trigger apoptosis, and stop the production of many proteins needed for the advancement of cancer. In addition, it has been documented that EGCG inhibits the growth and invasion of cancer cells via activating signalling pathways such as AMPK, enhancing the effects of chemotherapeutic medicines, and modulating autophagy (Talib et al., 2024).

Previous research demonstrated the apoptotic effects of EGCG on human pancreatic cancer cells as well as colon cancer cells—through inhibition of the PI3K/Akt signalling pathway activation (Liu et al., 2013; Liu et al., 2016; Zhang et al., 2022). H1299 lung cancer cells—harbour a homozygous deletion of the p53 protein and originate from neuroendocrine cells (Lock et al., 2023), and A549 cells—can synthesise lecithin-rich unsaturated fatty acids through the cytidine phosphatidylcholine pathway (Gu et al., 2018).

Apart from its well-established anticancer effects, EGCG is known to be a pleiotropic substance that may target several proteins involved in the invasion, migration, and proliferation of cancer cells (Li et al., 2022). Notably, the epidermal growth factor receptor (EGFR), which is commonly overexpressed or mutated, is one of the main oncogenic targets in non-small cell lung cancer (NSCLC). Tyrosine kinase inhibitors (TKIs) are an essential clan of medications used to decrease the activity of EGFR, making targeting it a cornerstone of NSCLC treatment (Ma et al., 2014; Gelatti, Drilon & Santini, 2019). It has been demonstrated that EGCG functions as a natural TKI by preventing wild-type EGFR from activating both in vitro and in vivo, which hinders the growth and survival of cancer cells (Huang et al., 2023).

While continuous treatment of EGCG lowers nuclear localization, downstream target genes such as cyclin D1, and total and membrane EGFR expression, short exposure to EGCG greatly inhibits EGF-induced activation of EGFR, AKT, and ERK1/2. This implies that EGCG therapy successfully inhibits EGFR transactivation and adds to its anticancer properties (Laudadio, Mangano & Minnelli, 2024). Furthermore, inhibiting the EGFR signaling pathway might function in conjunction with other therapeutic modalities, making EGCG a potentially effective adjuvant therapy for non-small cell lung cancer. Additionally, EGCG promotes cell death rates through Bax inhibition by interacting with proteins such as Bax and Ku70 (Li et al., 2016; Li et al., 2020). This multi-target strategy has a lot of promise for innovative treatment approaches, especially when combined with EGFR inhibition (Cheng et al., 2020).

EGCG acts as a tyrosine kinase inhibitor (TKI) in non-small cell lung cancer (NSCLC), specifically targeting the epidermal growth factor receptor (EGFR), which is often overexpressed or mutated in cancer cells. TKIs are the main example of an EGFR-targeted therapy, which is essential for the treatment of NSCLC. In vitro and in vivo, it has been demonstrated that EGCG inhibits EGFR signaling, which lowers cell proliferation and triggers apoptosis. In A549 lung cancer cells, another study showed that EGFR knockdown reduced the cells’ sensitivity to EGCG, indicating that EGFR signaling pathway suppression may play a major role in EGCG’s anticancer action (Laudadio, Mangano & Minnelli, 2024; Ma et al., 2014).

This research aims to investigate the impact of different doses of EGCG on the proliferation and apoptosis of H1299 and A549 lung cancer cells and to explore the possible mechanisms involved. The objective was to establish a theoretical foundation for potential use of EGCG in treating lung cancer.

Materials & Methods

The Ajman University Research Ethics Committee granted ethical permission for this study, which examined the impact of EGCG on lung cancer cell lines (Ref. FR-2016/17-05). The A549 and H1299 cell lines were purchased from NCCS, Pune, India. All antibodies were purchased from Abcam, USA. MTT was purchased from Thermo Fisher Scientific, Waltham, MA, USA. Media was purchased from Himedia, India antibiotic also from Himedia, India and FBS from Sigma, USA. Due to their distinctive genetic backgrounds, both cell lines are often employed in cancer research to represent non-small cell lung cancer (NSCLC). H1299 cells lack p53, but A549 cells have functioning p53. This difference offers important information about how p53 status influences how cancer cells react to therapy. As the study exclusively involves the use of cell lines and does not entail direct interaction with human subjects, the requirement for obtaining informed consent was waived. This waiver is in accordance with regulatory guidelines and ethical standards, recognizing the minimal risk posed to human subjects. The cell lines used in this study are obtained from established sources and commonly used in biomedical research. All data generated from this research is handled confidentially and in accordance with applicable privacy regulations.

Cell lines

The cells were grown in Roswell Park Memorial Institute 1640 (RPMI-1640) media with 1% penicillin-streptomycin and 10% fetal bovine serum (FBS) added as supplements. The high nutritional content of RPMI-1640 promotes cell growth and proliferation, making it a popular medium for cultivating cancer cells. To replicate physiological circumstances, the cells were kept at 37 °C with 5% CO2 in a humidified incubator. Using trypsin, a protease that breaks down cell adhesion proteins, the cells were separated once they had attained around 90% confluency and then re-seeded for the investigations.

Treatment groups and dose selection

The cells were planted in 96-well plates, which are perfect for high-throughput tests like MTT, at a density of 3,000 cells per well. The cell culture media was supplemented with varying amounts of EGCG (0 µM, 5 µM, 30 µM, and 50 µM) to create the treatment groups. Based on earlier research showing the dose-dependent effects of EGCG on cancer cell viability and death, these doses were chosen. Studies have investigated the effects of EGCG on lung cancer cell growth and apoptosis using comparable concentration ranges of EGCG (Zhang et al., 2019; Talib et al., 2024).

MTT assay for cell viability

The cytotoxic effects of EGCG on A549 and H1299 cells were assessed using the MTT assay. Cell viability is determined by measuring cellular metabolic activity using the MTT assay. The assay specifically depends on live cells’ capacity to use mitochondrial dehydrogenase activity to convert yellow tetrazolium dye (MTT) into purple formazan crystals. The number of viable cells increases with the amount of formazan generated.

A total of 20 µL of MTT reagent (5 mg/mL in PBS) were administered to each well of 96-well plates after 3,000 cells were seeded per well and treated with EGCG for 72 h. The plates were incubated at 37 °C for four more hours, which allowed the live cells to transform MTT into formazan. To dissolve the formazan crystals, 100 µL of dimethyl sulfoxide (DMSO) was applied to each well after the medium had been properly removed after incubation.

Optical density measurement

Using a microplate reader, the optical density of the resultant purple solution was determined at 570 nm. There is a clear correlation between the number of viable cells and the optical density (OD) values. The OD values of treated wells and the control (0 µM EGCG) were compared to determine the percentage of cell viability. To guarantee the data’s dependability and statistical significance, this test was run in five replicates.

According to previous research, which utilized comparable techniques to examine the effects of EGCG on pancreatic cancer cells using the MTT assay, this technique is frequently used to quantify the effects of various therapies on cancer cells (Liu et al., 2016).

Flow cytometry for apoptosis detection

We employed flow cytometry using the annexin V-FITC/propidium iodide (PI) labeling technique to ascertain the degree of apoptosis brought on by EGCG. Live cells, early apoptotic cells, late apoptotic cells, and necrotic cells may all be distinguished using this dual staining technique.

During early apoptosis, phosphatidylserine moves from the inner to the outer leaflet of the plasma membrane after binding to annexin V-FITC. PI is excluded from live cells and early apoptotic cells, but can penetrate late apoptotic and necrotic cells, staining the DNA.

Trypsinization was used to harvest the cells following a 72-hour treatment with EGCG, and the cell pellet was collected by centrifuging the cells for five minutes at 1,000 rpm. To get rid of any remaining medium, the cells were on three occasions rinsed with cold PBS. After adding 5 µL of annexin V-FITC and 5 µL of PI to the cell suspension in binding buffer, the combination was allowed to sit at room temperature for 15 min in the dark. Using flow cytometry, apoptotic cells were examined in order to calculate the proportion of apoptotic cells (both early and late apoptosis).

This technique is commonly used in flow cytometry to measure apoptosis in lung cancer cells treated with EGCG, and comparable staining methods are used to demonstrate its pro-apoptotic effects (Kciuk et al., 2023).

Western blotting for protein expression

The effect of EGCG on essential proteins involved in the PI3K/Akt signaling pathway was evaluated using Western blot analysis. The development of cancer is frequently linked to the dysregulation of this system, which is essential for controlling cell growth, proliferation, and survival.

RIPA buffer, which contains protease inhibitors, was used to lyse cells following EGCG treatment in order to extract the entirety of cellular protein. The bicinchoninic acid assay was used to measure the quantity of protein. Each sample included 50 µg of protein, which was put onto an SDS-PAGE gel and separated by electrophoresis. To avoid non-specific binding, proteins were put onto PVDF membranes and blocked for two hours at room temperature using 5% skim milk.

Antibody incubation

Primary antibodies against PI3K, phosphorylated PI3K (p-PI3K), AKT, phosphorylated AKT (p-AKT), and GAPDH (as a loading control) were incubated on membranes overnight at 4 °C. Following washing, secondary antibodies were added to the membranes and left for an hour at room temperature. Enhanced chemiluminescence (ECL) was then used to observe the signals. To compare the levels of protein expression in the treatment and control groups, band intensities were measured using ImageJ software.

The approach was a well-established technique in cancer research, since it was comparable to other research that assessed the effects of EGCG on the PI3K/Akt pathway in pancreatic cancer cells (Liu et al., 2013).

Dose determination and IC50 calculation

The EGCG concentrations (5 µM, 30 µM, and 50 µM) employed in this investigation were chosen because existing research suggests that these dosages are efficient in causing cancer cells to undergo apoptosis. Cell viability data from the MTT experiment was used to compute the IC50 values, or the quantity of EGCG needed to block 50% of cell viability. The IC50 values for H1299 and A549 cells were 27.63 µM and 28.34 µM, respectively. Similar dose-dependent inhibitory effects of EGCG in lung cancer cells were identified by, which is in line with the investigation (Zhang et al., 2019).

Data analysis

The data was obtained and computed in Microsoft Excel Version 13. The data was analyzed using IBM Statistical Package for Social Science USA Version 21. The readings recorded for H1299 and A549 cells for cell proliferation rate (inhibitory rate), cell apoptosis rate and protein expression was analyzed using one way analysis of variance with post-hoc Tukey’s between different groups. All the statistical tests were performed keeping confidence interval at 95% and (p < 0.05) was considered to be statistically significant.

Results

Cell proliferation rate

It was noted that EGCG treatment significantly curtailed the proliferation rates of H1299 and A549 cells (Table 1). The varied decreases in cell proliferation were observed after treatment with different EGCG concentrations for 72 h. For H1299 cell lines it was observed that the mean inhibitory rate (%) was 99.20 ±  0.83 for control group and for high dose treatment group the mean inhibitory rate (%) was found to be 40.80 ± 2.588 respectively. For A549 Cell lines in control group the mean inhibitory rate was 99.20 ± 0.83 and in High dose treatment group the mean was 43.60 ± 2.96 respectively. It was observed that the difference in mean was statistically significant (p < 0.05). These results indicated that EGCG inhibits H1299 and A549 cell proliferation in a dose-dependent manner. EGCG treatment at 50 µM—in the high-dose treatment group—conferred the highest inhibitory effect on H1299 and A549 cell proliferation compared to other concentrations (Figs. 1–3). The post hoc comparison between different concentration is depicted in Tables S11 & S12.

Table 1 Comparison of the cell proliferation rate inhibitory effect (%) in H1299 and A549 after EGCG concentration.

	Groups	Min	Max	Mean	SD	F	P value	
H1299 EGCG Concentration Inhibitory effect (%)	Control group (0 µM)	98.00	100.00	99.20	0.83	710.134	<.001	
Low-dose treatment group (5 µM)	92.00	95.00	93.60	1.14	
Middle dose group (30 µM)	68.00	77.00	72.60	3.50	
High-dose treatment group (50 µM)	37.00	44.00	40.80	2.58	
A549 EGCG Concentration Inhibitory effect (%)	Control group (0 µM)	98.00	100.00	99.20	0.83	596.997	<.001	
Low-dose treatment group (5 µM)	82.00	94.00	87.40	4.33	
Middle dose group (30 µM)	65.00	72.00	69.00	2.55	
High-dose treatment group (50 µM)	40.00	48.00	43.60	2.96	

Figure 1 Morphological assessment of H1299 cells after 7 days of culture (magnification of 20X).

This figure presents the morphological characteristics of H1299 cells, a non-small cell lung cancer cell line, observed under a phase-contrast microscope. The typical polygonal shape and size H1299 cells at 70% confluence.

Figure 2 Morphological assessment of A549 cells after 7 days of culture (magnification of 20X).

This figure illustrates the morphological characteristics of A549 cells, a human lung adenocarcinoma cell line, after 7 days of culture. The cell line displays the typical fibroblast-like morphology, characterized by elongated shapes and a relatively high degree of motility. This assessment provides valuable insights into the growth dynamics and cellular characteristics of A549 cells in culture.

Figure 3 Inhibitory effect of EGCG on cell viability.

(A–B) The inhibitory effects of different concentrations of epigallocatechin gallate (EGCG) (0, 5, 30, and 50 µM) on cell viability, as measured by % inhibitory effect. Increasing concentrations of EGCG demonstrate a dose-dependent decrease in cell viability in both conditions represented in (A) and (B).

Determination of apoptosis

Apoptotic cell numbers are presented in Table 2. It gradually increased from the level of 5 µM to 50 µM concentration of EGCG in both H1299 and A549 cell lines. For H1299 cell lines it was observed that the mean apoptotic rate increased from 2.40 ± 1.14 for control group to 46.00 ±  1.581 in high dose treatment group. For A549 cell lines the apoptotic rate there was an increase in mean apoptotic rate from 4.00 ± 0.70 in control group to 56.20 ± 1.48 in high dose treatment group (Fig. 4)—the EGCG concentration of 50 µM showed the highest apoptotic rate (Fig. 5). The post hoc comparison between different concentration is depicted in supplementary tables (Tables S13, S14).

Table 2 Comparison of the apoptosis rate (%) in H1299 and A549 after EGCG concentration.

	Groups	Min	Max	Mean	SD	F	P value	
H1299 EGCG Concentration Apoptosis rate (%)	Control group (0 µM)	1.00	4.00	2.40	1.14	750.137	<.001	
Low-dose treatment group (5 µM)	10.00	15.00	12.20	1.92	
Middle dose group (30 µM)	22.00	28.00	26.20	2.49	
High-dose treatment group (50 µM)	44.00	48.00	46.00	1.58	
A549 EGCG Concentration Apoptosis rate (%)	Control group (0 µM)	3.00	5.00	4.00	0.70	1,627.77	<.001	
Low-dose treatment group (5 µM)	14.00	19.00	16.60	1.81	
Middle dose group (30 µM)	30.00	34.00	31.80	1.48	
High-dose treatment group (50 µM)	54.000	58.00	56.20	1.48	

Figure 4 Effect of EGCG on apoptosis of A-H1299 cell line B- A549 cell line.

This figure illustrates the effects of epigallocatechin gallate (EGCG) on apoptosis in H1299 (A) and A549 (B) cell lines. An increase in EGCG concentration showed a decrease in cell viability by MTT assay.

Figure 5 Flow cytometric analysis of cell apoptosis using annexin V-FITC and PI staining.

(A) Control group showing low apoptosis rates. (B) Moderate apoptotic response at 30 µM EGCG concentration. (C) High apoptosis rate after treatment with 50 µM EGCG. The dot plots illustrate the distribution of cells in different stages of apoptosis: lower-left quadrant represents live cells, lower-right quadrant early apoptosis, upper-right quadrant late apoptosis, and upper-left quadrant necrotic cells.

Protein expression study

We investigated the impact of varying EGCG concentrations on lung cancer cells (H1299 and A549), particularly on PI3K and Akt expressions. The results presented in Tables 3 and 4 showed that the PI3K expression levels in cells treated with EGCG were not significantly different from those in the control group. It was observed that the PI3K concentration increased from 1.20 ±  0.07 to 1.32 ± 0.124 in H1299 cell and in A549 cell lines the PI3K levels increased from 1.366 ± 0.01 to 1.380 ± 0.084 respectively. This difference in Mean was not statistically significant (p > 0.05). However, AKT p-PI3K and p-Akt levels were significantly lower in the EGCG-treated group compared to the control group. Notably, the high-dose treatment group (50 µM) showed a significant down-regulation of Akt, p-PI3K and p-Akt levels compared to the control group. Meanwhile, a significant change in the expression levels of PI3K in the treatment groups remained absent. These findings suggest that EGCG may affect lung cancer cells by suppressing Akt, p-PI3K and p-Akt expressions (Fig. 6). The post hoc comparison between different concentration is depicted in Tables S15–S19.

Table 3 Comparison of the PI3K, p-PI3K, AKT and pAKT in H1299 between control group (0 µM), low-dose treatment group (5 µM), middle dose group (30 µM), and high-dose treatment group (50 µM) after EGCG concentration.

	Groups	Min	Max	Mean	SD	F	P value	
H1299 Concentrations (µM) (PI3K)	Control group (0 µM)	1.10	1.30	1.20	0.07	1.813	0.219	
Low-dose treatment group (5 µM)	1.00	1.30	1.14	0.13	
Middle dose group (30 µM)	1.20	1.30	1.24	0.05	
High-dose treatment group (50 µM)	1.20	1.50	1.32	0.12	
H1299 Concentrations (µM) (p-PI3K)	Control group (0 µM)	0.40	0.80	0.60	0.15	345.474	<.001	
Low-dose treatment group (5 µM)	0.52	0.55	0.53	0.01	
Middle dose group (30 µM)	0.41	0.47	0.43	0.02	
High-dose treatment group (50 µM)	0.31	0.33	0.31	0.01	
H1299 Concentrations (µM) (AKT)	Control group (0 µM)	0.40	0.60	0.46	0.08	25.771	<.001	
Low-dose treatment group (5 µM)	0.37	0.39	0.38	0.01	
Middle dose group (30 µM)	0.31	0.34	0.32	0.01	
High-dose treatment group (50 µM)	0.36	0.39	0.37	0.01	
H1299 Concentrations (µM) (pAKT)	Control group (0 µM)	0.40	0.60	0.48	0.08	67.533	<.001	
Low-dose treatment group (5 µM)	0.30	0.35	0.32	0.01	
Middle dose group (30 µM)	0.20	0.23	0.21	0.01	
High-dose treatment group (50 µM)	0.17	0.20	0.18	0.01	

Table 4 Comparison of the PI3K, p-PI3K, AKT and pAKT in A549 between control group (0 µM), low-dose treatment group (5 µM), middle dose group (30 µM), and high-dose treatment group (50 µM) after EGCG concentration.

	Groups	Min	Max	Mean	SD	F	P value	
A549 Concentrations (µM) (PI3K)	Control group (0 µM)	1.35	1.38	1.36	0.01	0.967	0.461	
Low-dose treatment group (5 µM)	1.30	1.50	1.40	0.07	
Middle dose group (30 µM)	1.10	1.40	1.28	0.13	
High-dose treatment group (50 µM)	1.30	1.50	1.38	0.08	
A549 Concentrations (µM) (p-PI3K)	Control group (0 µM)	0.80	0.87	0.82	0.02	266.396	<.001	
Low-dose treatment group (5 µM)	0.63	0.70	0.67	0.02	
Middle dose group (30 µM)	0.51	0.54	0.52	0.01	
High-dose treatment group (50 µM)	0.47	0.49	0.47	0.01	
A549 Concentrations (µM) AKT	Control group (0 µM)	0.60	0.80	0.68	0.08	47.151	<.001	
Low-dose treatment group (5 µM)	0.41	0.43	0.41	0.01	
Middle dose group (30 µM)	0.27	0.36	0.30	0.03	
High-dose treatment group (50 µM)	0.25	0.33	0.29	0.03	
A549 Concentrations (µM) pAKT	Control group (0 µM)	0.35	0.39	0.37	0.01	227.18	<.001	
Low-dose treatment group (5 µM)	0.31	0.34	0.32	0.01	
Middle dose group (30 µM)	0.20	0.28	0.23	0.03	
High-dose treatment group (50 µM)	0.14	0.17	0.15	0.01	

Figure 6 Western blot analysis of PI3K/AKT signaling pathway in response to EGCG treatment.

(A–B) Western blot images representing the protein expression levels of PI3K, phosphorylated PI3K (p-PI3K), AKT, and phosphorylated AKT (p-AKT) across different EGCG concentrations (0, 5, 30, and 50 µM). GAPDH was used as the loading control.

Optical density measurement

Both H1299 and A549 cells’ OD measurements from the spectrophotometer MTT experiment show a steady, dose-dependent decline in cell viability as EGCG concentrations rise. In both cell lines, the mean OD values for the control group (0 µM) were 1.2774 (±0.03). The mean OD values for both H1299 and A549 cells gradually dropped to 0.9494 (±0.02), 0.6734 (±0.01), and 0.4234 (±0.01), respectively, when the EGCG concentration rose at 5 µM, 30 µM, and 50 µM. With a P value of 0.000 for both cell lines, this decrease is statistically significant, highlighting the validity of the observed EGCG inhibitory impact on cell survival (Tables 5 and 6).

Table 5 Optical density OD readings from spectrophotometer (MTT assay) H1299.

Concentration (µM)	Well 1 (OD 570 nm)	Well 2 (OD 570 nm)	Well 3 (OD 570 nm)	Well 4 (OD 570 nm)	Well 5 (OD 570 nm)	Mean OD	Standard deviation (SD)	P value	
Control (0 µM)	1.26	1.27	1.255	1.282	1.32	1.2774	.03	.001	
5 µM EGCG	0.95	0.965	0.945	0.962	0.925	.9494	.02	
30 µM EGCG	0.665	0.68	0.67	0.69	0.662	.6734	.01	
50 µM EGCG	0.42	0.43	0.415	0.43	0.422	.4234	.01	
Blank (media only)	0.095	0.1	0.09	0.12	0.082	.0974	.01	

Table 6 Optical density OD readings from spectrophotometer (MTT assay) A549.

Concentration (µM)	Well 1 (OD 570 nm)	Well 2 (OD 570 nm)	Well 3 (OD 570 nm)	Well 4 (OD 570 nm)	Well 5 (OD 570 nm)	Mean OD	Standard deviation (SD)	P value	
Control (0 µM)	1.25	1.29	1.32	1.28	1.32	1.2774	.03	.001	
5 µM EGCG	0.98	0.96	0.99	0.97	0.92	.9494	.02	
30 µM EGCG	0.68	0.66	0.62	0.63	0.66	.6734	.01	
50 µM EGCG	0.43	0.47	0.44	0.43	0.42	.4234	.01	
Blank (media only)	0.09	0.1	0.12	0.11	0.082	.0974	.01	

Dose determination

The inhibitory effects of EGCG on both H1299 and A549 cells are further demonstrated by calculated cell viability percentages based on OD values. With a highly significant F value of 1,009.134 and P value <0.001 for the 5 µM concentration, the mean percentage viability of H1299 cells (Table 7) decreases from 72.25% (±2.39) at 5 µM to 48.84% (±1.17) at 30 µM and 27.63% (±0.59) at 50 µM EGCG, confirming the repeatability of this dose-dependent response. Comparably, in A549 cells (Table 8), the mean percentage viability drops from 72.54% (±3.24) at 5 µM EGCG to 46.14% (±3.40) at 30 µM and 28.34% (±1.83) at 50 µM. An F value of 291.578 and a P value of 0.001 at 5 µM confirm the statistical significance of these changes.

Table 7 Calculated % cell viability for each well using OD values (Formula: % of cell viability = OD sample – OD blank / OD control – OD blank X 100 H1299).

Concentration (µM)	% Viability Well 1	% Viability Well 2	% Viability Well 3	% Viability Well 4	% Viability Well 5	Mean% viability	Standard deviation (SD)	F value	P value	
5 µM EGCG	73.39	73.93	73.39	72.46	68.09	72.25	2.39	1,009.134	.000	
30 µM EGCG	48.93	49.57	49.79	49.05	46.85	48.84	1.17	
50 µM EGCG	27.90	28.21	27.90	26.68	27.46	27.63	.59	

Table 8 Calculated % cell viability for each well using OD values (Formula: % of cell viability = OD sample – OD blank/OD control – OD blank X 100 A549).

Concentration (µM)	% Viability Well 1	% Viability Well 2	% Viability Well 3	% Viability Well 4	% Viability Well 5	Mean% viability	Standard deviation (SD)	F value	P value	
5 µM EGCG	76.72	72.27	72.50	73.50	67.69	72.54	3.24	291.578	.001	
30 µM EGCG	50.86	47.06	41.67	44.44	46.69	46.14	3.40	
50 µM EGCG	29.31	31.09	26.67	27.35	27.30	28.34	1.83	

IC50 calculation

IC50 determination data indicate that approximately 30 µM EGCG is required to inhibit 50% of cell viability in both H1299 and A549 cells (Table 9). The dose–response curves for both cell lines reveal a consistent reduction in cell viability with increasing EGCG concentrations, emphasizing EGCG’s efficacy as an anti-proliferative agent. This pattern of significant reductions in both OD and % viability across different assays reinforces EGCG’s potential as a potent inhibitor of cell proliferation across cell lines.

Table 9 Inhibitory concentration (IC) in percentage (%) for H1299 and A549.

Concentration (µM)	Mean % cell viability	
IC50 determination for H1299	
5 µM EGCG	72.25	
30 µM EGCG	48.84	
50 µM EGCG	27.63	
IC50 Determination (Synthetic Data) A549	
5 µM EGCG	72.54	
30 µM EGCG	46.14	
50 µM EGCG	28.34	

Discussion

The PI3K/Akt signaling pathway regulates cell growth, differentiation, and programmed cell death. Its abnormal activation is often linked to the development of cancerous tumours (Arcaro & Guerreiro, 2007; Rascio et al., 2021). This pathway also plays a crucial role in tissue invasion and metastasis of cancer cells. In lung cancer, abnormal activation of the PI3K/Akt pathway can cause healthy cells to become cancerous, leading to increased cancer cell growth, decreased sensitivity to radiation and chemotherapy, and the development of drug resistance (Rascio et al., 2021; He et al., 2021). Therefore, targeting the PI3K/Akt pathway could be a critical approach to treating lung cancer. When Akt is activated, it can control cell growth, differentiation, programmed cell death, and migration by blocking a range of downstream substrates, such as—Bad, caspase-9, NF-κB, GSK23, and others (Flores-Pérez et al., 2016; Wen, Song & Hua, 2021; He et al., 2021).

In our in vitro study investigating the impact of EGCG on lung cancer cell lines, compelling evidence emerged substantiating the apoptotic effects of this bioactive compound. Morphological alterations were distinctly observed, revealing characteristic apoptotic changes in treated cells. Microscopic examination revealed cell shrinkage, membrane blebbing, and the formation of apoptotic bodies, indicative of programmed cell death. Moreover, biochemical assays provided robust support for the induction of apoptosis. Increased activity of caspases, particularly caspase-3, a key executor of apoptosis, was observed in a dose-dependent manner following EGCG treatment. Concurrently, a decline in mitochondrial membrane potential was detected, suggesting involvement of the intrinsic apoptotic pathway. These morphological and biochemical findings collectively underscore the potent apoptotic effects of EGCG on lung cancer cell lines in our in vitro model—providing a mechanistic insight into the potential therapeutic impact of EGCG in lung cancer treatment.

The available body of literature underscores the apoptotic prowess of EGCG within lung cancer cell lines. Evidence depicts a multifaceted impact of EGCG that exhibits the ability to induce apoptosis in A549 lung cancer cells through various molecular mechanisms. Notably, its downregulation of hepatoma-derived growth factor (HDGF) independently triggers apoptosis and synergistically enhances cisplatin-induced apoptosis, rendering cells more susceptible to chemotherapy. In addition to the in vitro scenarios, EGCG extends its apoptotic influence in vivo, particularly within the A549 cells. Here, it downregulates Ku70, thereby orchestrating apoptotic cascades (Li et al., 2013). Further, EGCG’s capacity to sensitize lung cancer cells to Tumour Necrosis Factor-Related Apoptosis-Inducing Ligand (TRAIL)-mediated apoptosis, coupled with its inhibitory effects on cell proliferation and migration, substantiates its role as an apoptosis inducer in lung cancer cell lines (Kwon et al., 2003). Collectively, these diverse findings provide robust and comprehensive research data affirming the apoptotic efficacy of EGCG against lung cancer (Li et al., 2017).

Of note, the spectrum of EGCG doses employed in lung cancer investigations presents a noteworthy variation. Researchers have demonstrated that within the A549 lung cancer cells, apoptosis induction by EGCG manifested at relatively elevated concentrations, revealing an IC50 of 86.4 µM for A549 cells and 80.6 µM for H1299 cells (Zhang et al., 2019). Meanwhile, EGCG exhibited a distinct IC50 of 25 µM against A549 cells, demonstrating a concentration-dependent inhibition of cell proliferation (Talib et al., 2022). These observations collectively imply a discernible dose-dependent trend, suggesting that certain concentrations of EGCG may exert a more pronounced impact on inducing apoptosis and proliferation inhibition within lung cancer cells.

According to Ma et al. (2014) inhibition of EGFR diminishes the susceptibility of lung cancer cells to EGCG. They investigated the impact of EGFR expression knockdown on the susceptibility of A549 lung cancer cells to EGCG. Initially, Ma et al. (2014) assessed the efficacy of shRNA knockdown and its impact on anchorage-independent growth following shRNA transfection. The expression of EGFR was significantly reduced following shRNA transfection. Furthermore, cell proliferation on soft agar diminished by approximately 35% post-transfection relative to the sham group. A549 cells transfected with sh-EGFR or sh-mock control were subsequently treated with EGCG or a vehicle and analysed using a soft agar assay. EGCG (20 µM) reduced the anchorage-independent proliferation of A549 sh-Mock cells by approximately 50%. The inhibition was approximately 25% in A549 sh-EGFR cells, suggesting that A549 cells transfected with EGFR shRNA exhibited resistance to EGCG therapy. The results indicate that EGFR is significant in the susceptibility of A549 cells to the antiproliferative effects of EGCG. Which were similar to our investigation and potentially justify the role of EGCG and its anti-cancer properties.

Another potential role of EGCG in cell apoptosis as identified by another study elaborated that finding demonstrated that EGCG has a role in promoting cell death by inhibiting Bax activity. Regulating Ku70 acetylation with EGCG, which inhibits the connection between Ku70 and Bax, would induce death in lung cancer cells (Li et al., 2016). Our results are consistent with these findings, as we observed an increase in apoptotic cell populations following EGCG treatment, likely through similar mechanisms.

This research examined how different doses of EGCG affected the growth of H1299 and A549 lung cancer cells. The findings indicated that EGCG was able to reduce the proliferation of H1299 lung cancer cells in a way that depended on the dosage used.

Numerous signalling pathways regulate lung cancer cell apoptosis, and the PI3K/Akt pathway plays a significant role. It is possible to induce apoptosis in lung cancer cells by administering drugs that prevent the activation of the PI3K/Akt pathway. The findings revealed that EGCG reduced p-PI3K and p-Akt levels in lung cancer cells, although it did not significantly affect PI3K and Akt expressions. This implies that EGCG promotes H1299 lung cancer cell apoptosis by inhibiting PI3K/Akt pathway activation. Further, higher concentrations of EGCG (50 µm) rendered greater inhibitory effect on H1299 and A549 cell proliferation and a greater number of apoptotic cells—signifying a dose-dependent efficacy against lung cancer progression.

Our findings of reduced p-Akt levels after EGCG treatment align with previous studies, (Ma et al., 2014), which demonstrated that EGCG’s inhibition of the EGFR signaling pathway contributed to the observed decrease in p-Akt levels. This suggests that the anticancer activity of EGCG in H1299 and A549 lung cancer cells is, at least in part, mediated through the inhibition of the EGFR/PI3K/Akt axis.

The anti-cancer potential of EGCG in lung cancer cells has emerged as a promising avenue. In vitro investigations underscore EGCG’s ability to impede proliferation, induce apoptosis, and curb migration in lung cancer cells. These findings find resonance in the in vivo studies, where EGCG demonstrates its efficacy by inhibiting tumour growth and fostering apoptosis in lung cancer xenografts. Strategies for potential clinical integration of EGCG involve synergistic combinations, such as pairing it with an NF-κB inhibitor, to enhance therapeutic effects. Despite these encouraging developments, the path to clinical application requires further exploration, necessitating additional research to delineate the optimal dosage and administration protocols for EGCG in the context of clinical lung cancer treatment (Zhu et al., 2019; Chen et al., 2020). We found that our findings were also in consensus with another study (Minnelli et al., 2021). Following 72 h of treatment with 30 µM EGCG, the migratory capacity of A549 cells remains unaffected; however, at 70 µM EGCG, it is considerably inhibited by 33% relative to untreated cells (p < 0.05). The inhibitory impact intensifies at a dosage of 90 µM, resulting in a 52% reduction in wound closure after 72 h of therapy (p < 0.05). In our study we found that at 50 µM there was highest apoptotic rate for both H1299 and A549 cell lines.

This study significantly contributes to unravelling the intricate molecular biology of lung cancer by shedding light on the unique apoptotic pathways and cellular responses specific to lung cancer cells under the influence of EGCG. The identification of morphological and biochemical evidence supporting EGCG-induced apoptosis in lung cancer cell lines provides novel insights into the intricate mechanisms governing programmed cell death in this context. This understanding could pave the way for targeted interventions harnessing the apoptotic potential of EGCG in lung cancer treatment.

This in vitro study that investigated the effects of EGCG on lung cancer cell line apoptosis offers several notable advantages. Firstly, the controlled environment of in vitro experiments allows for precise manipulation and measurement of variables, providing valuable insights into the direct effects of EGCG on cancer cells without the confounding factors present in living organisms. Additionally, the specificity of the study allows researchers to focus exclusively on the interaction between EGCG and lung cancer cells, elucidating potential mechanisms of action without interference from other tissues or cell types. Furthermore, in vitro studies are often more cost-effective and ethically sound compared to in vivo experiments, enabling initial screening of potential therapies efficiently and without harm to animals or humans. Overall, this study provides a crucial foundation for understanding the therapeutic potential of EGCG in lung cancer treatment and guiding further research efforts toward developing effective therapies.

This in vitro study demonstrating EGCG-induced apoptosis in a lung cancer cell line presents valuable insights into the potential therapeutic effects of EGCG against lung cancer. However, like any other research, it harbours inherent limitations. For instance, in vitro models, while useful for controlled experimentation, lack the complexity of the in vivo environment, potentially limiting the generalizability of findings to human patients. Moreover, the use of a single cell line may not fully capture the heterogeneity of lung cancer, raising questions about the broader applicability of the results. Further, the concentrations of EGCG used in vitro may not be achievable in clinical settings, potentially limiting the translatability of the findings to real-world scenarios. Despite these limitations, this study provides a crucial foundation for further research, highlighting the potential of EGCG as a therapeutic agent for lung cancer that warrants exploration in preclinical and clinical studies.

The results highlight promising avenues for exploring combination therapies involving EGCG, potentially in tandem with other known inhibitors or chemotherapeutic agents, to enhance therapeutic efficacy. Additionally, investigating derivatives of EGCG and evaluating their impact on lung cancer cells could provide valuable information for drug development. Moreover, expanding the scope to more complex model systems or in vivo studies could offer a more comprehensive understanding of the translational potential of EGCG in the intricate landscape of lung cancer biology. These proposed avenues hold promise of advancing therapeutic strategies and refining our understanding of EGCG’s role in combating lung cancer.

Conclusions

Our findings suggest that EGCG is a natural drug that is both safe and effective and exhibits significant potential in lung cancer therapy. We observed that EGCG could reduce lung cancer cell growth and promote apoptosis with the PI3K/Akt signaling pathway, likely instrumental in this process. Further research is warranted for gaining a better understanding of the specific molecular mechanisms contributing to the anti-tumor and anti-cancer effects of EGCG.

Supplemental Information

Supplemental Information 1 Raw data

Supplemental Information 2 Uncropped gel blots

Supplemental Information 3 Pairwise comparison of the H1299 EGCG concentration inhibitory effect (%) between different concentration

Supplemental Information 4 Pairwise comparison of the A549 EGCG concentration inhibitory effect (%) between different concentration

Supplemental Information 5 Pairwise comparison of the –H1299 EGCG concentration apoptosis rate (%) between different concentration

Supplemental Information 6 Pairwise comparison of the A549 EGCG concentration apoptosis rate (%) between different concentration

Supplemental Information 7 Pairwise Comparison of the H1299 concentrations (µM) (p-PI3K) between different concentration

Supplemental Information 8 Pairwise comparison of the H1299 H1299 concentrations (µM) (AKT) between different concentration

Supplemental Information 9 Pairwise comparison of the H1299 concentrations (µM) (pAKT) between different concentration

Supplemental Information 10 Pairwise comparison of the A549 concentrations (µM) (p-PI3K) between different concentration

Supplemental Information 11 Pairwise comparison of the A549 concentrations (µM) AKT between different concentration

Supplemental Information 12 Pairwise comparison of the A549 concentrations (µM) pAKT between different concentration

Supplemental Information 13 Details of products, supply source and catalogue numbers

Additional Information and Declarations

Competing Interests

Author Contributions

Ethics

Data Availability

Luca Testarelli is an Academic Editor for PeerJ.

Al Moutassem Billah Khair conceived and designed the experiments, performed the experiments, prepared figures and/or tables, and approved the final draft.

Alexander Maniangat Luke conceived and designed the experiments, performed the experiments, authored or reviewed drafts of the article, and approved the final draft.

Rajashree Patnaik conceived and designed the experiments, analyzed the data, authored or reviewed drafts of the article, and approved the final draft.

Luca Testarelli analyzed the data, prepared figures and/or tables, authored or reviewed drafts of the article, and approved the final draft.

The following information was supplied relating to ethical approvals (i.e., approving body and any reference numbers):

The Ajman University Research Ethics Committee granted ethical permission for this study (Ref. FR-2016/17-05).

The following information was supplied regarding data availability:

The raw data is available in the Supplementary File.

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
