# Peer review of "EGCG’s anticancer potential unveiled: triggering apoptosis in lung cancer cell lines through in vitro investigation"

_PeerJ, doi:10.7717/peerj.19135_

## Round 0.1 · original submission · Major Revisions

The current manuscript would benefit from improvements in several key areas namely, performing statistical analysis, sample size, and providing a more comprehensive discussion of the reported results. Crucial details necessary for replication by other researchers are currently lacking. Additionaly, the introduction requires revision to ensure relevance, particularly regarding the transition from current treatment strategies to EGCG. Also, where the cells seeded at low or high density? This information would provide further details on why the cell morphology do not look healthy at 7 days of culture from the light microscopy images provided. Lastly, please see the comments provided by the reviewers for additional clarification.

·

Basic reporting

I commend the authors for their clear English used throughout the manuscript. In addition, the structure conforms to PeerJ standards. The introduction and background show context with referenced literature. However, there is a weakness in the figures which are not properly labelled or described and need more detail to meet publication standards. I suggest that the authors use detailed legends below the figures to provide more information on the figures, instead of at the top.
Raw data has been shared, but not sufficient enough for some methods described. The correct provenance of the A549 and H1299 cell lines is described.

Experimental design

Original primary research within the scope of PeerJ. Research question is well defined and meaningful, and research fills an identified knowledge gap.
In addition, ethical standards are high and rigorous. However, some methods are not described with sufficient details or information for reproducibility while others not rigorously investigated.
Please, authors should provide detailed information on MTT assay, Flow cytometry, Western Blot (see general comments below).

Validity of the findings

Thank you for providing the underlying data, however not all have been provided. Raw data for original OD value readings from the spectrophotometer are not provided. Also, raw data after calculating %cell viability for each well using the OD values and formular (on line 145) are not provided. Data and figures on how IC50 was obtained are not supplied. Please authors should provide all relevant data for easier and better comprehension.

Data provided was not statistically analyzed in this study. To make the results more meaningful, I’ll suggest you determine if the differences observed between the different groups is statistically significant using p-values (specifically, Figures 3 & 4).

Figures 1 and 2 equally have no legends and the objective of the images (X10, X40, X100) need to be included.
Figure 5 would only be meaningful, if the image legends provide more detail. Improvements can be done by describing the change in cell morphology observed as well as explaining what each quadrant represents

Since statistical analysis were not done in this study, how do you measure levels of significance? In lines 188, 244 and 298 you mention significant difference in the effect and potential of EGCG whereas this cannot be determined visually.

Conclusions are well stated, linked to the original research question and limited to supporting results.

Additional comments

General comments
The paper investigates the anti-cancer potential of different doses of ECGC on the proliferation and inducing apoptosis in lung cancer cell lines A549 and H1299 in vitro. The authors claim that ECGC is safe, effective and exhibits significant potential in lung cancer therapy by reducing cell growth and promoting apoptosis with the PI3K/Akt signaling pathway. Thus, promising as a natural therapeutic agent in the treatment of lung cancer.

However, I also have a few comments and questions by section for clarity and improved quality of the paper.

Abstract
Your abstract lacks coherence with regards to the cell lines used in this study. the background and methods make mention of H1299 cells only, while the results report both H1299 and A549 cells lines. This does not add up. It seems like the experiments were not performed on both cell lines, which is not reflective of the study. Please correct this for clarity and coherence in the abstract.
Again, your abstract has no keywords. Please authors need to include keywords as this is essential for any paper.

Materials and Methods
Although the correct provenance of cell lines is described, in your materials section, only the cell lines used in the study are mentioned. I suggest you equally mention the reagents/kits as well (specifically, you should indicate the antibodies, their catalogue # and where they were purchased/obtained from.

Please can you explain how you arrived at the different dose concentrations or treatment groups (0uM, 5uM, 30uM, 50uM)? Did the authors determine the effect of ECGC on both cell lines? What was the IC50 of ECGC on A549 and H1299? There is no data or figure showing this? Please include data on this if it was done.

For the MTT dye reduction test, there is no mention of the cell density seeded per well at the beginning of this MTT assay, which is crucial. I suggest that you improve the description of the assay at line 136-138 by mentioning the cell density seeded per well at the beginning of the experiment as well as line 139 by mentioning the initial color of the tetrazolium dye.

Again, it is questionable why you performed the MTT assay in a 6 well plate instead of the recommended 96 well microplate or 384 wells (line 137). Did you have a positive and negative control using the 6well plate? This makes it impossible to obtain OD readings in replicates. And also impossible to determine Mean OD values for each treatment condition. I strongly suggest that this assay be re-done using either a 96 well plate or 384 wells and all the raw data provided in the supplementary files.

Detection of apoptosis. If cells are centrifuged and collected, then it is important to mention the speed and duration. Please include this detail on line 151. On line 152, the Annexin V apoptosis detection kit’s working concentration used in the experiment is not known. At what concentration was it added to the cells?
Please can the authors explain what FITC and Propidium Iodide in the images represent? (Figure 5)

Western blotting. Please what was the composition of your lysis buffer used? Line 160. This has to be mentioned as well. In addition, how long was the secondary antibody added to the membrane? And under what conditions? Was it overnight at 4C? At room temperature?

Statistical Analysis
How were your results analyzed? Data not statistically analyzed. It would be good if the authors could statistically analyze their data to effectively compare the differences observed in the different group treatments and to know if these differences are significant or not.

Results
Are relevant to the hypothesis though some are not well represented.

Discussions
In as much as authors seek to interprete results in context of existing literature as well as compare their findings with previous studies, in this study, they are more focused on discussing others findings rather than the findings of this study.

Reviewer 2 ·

Basic reporting

In the introduction section, authors described the already known anticancer properties of EGCG. However, there are some points relevant for their studies that need to be discussed. EGCG is a pleiotropic molecule able to target several proteins involved in cancer cell proliferation, invasion, and migration. In NSCLC, one of the main oncogenic targets is the epidermal growth factor receptor (EGFR) which is frequently overexpressed or mutated. In this context, EGFR-targeted therapy is mainly represented by tyrosine kinase inhibitors (TKIs) (reviewed in doi.org/10.1021/acschembio.4c00028) and neither information is reported about this. It is also known that EGCG acts as a TKI by targeting EGFR wild type both in vitro and in vivo.

The authors’ research aims to investigate the impact of EGCG on proliferation and apoptosis in H1299 and A549 lung cancer cells to explore the possible mechanisms involved. Yu-Chao Ma et al, showed that the knockdown of EGFR in lung cancer cells (including A549) decreased their sensitivity to EGCG. Thus, inhibition of the EGFR signaling pathway may partially contribute to the anticancer activity of EGCG (doi.org/10.3892/or.2013.2933). And this could explain also the reduction expression levels of phosphorylated Akt (p-Akt) induced by EGCG that the authors have demonstrated in this work. Moreover, EGCG induces also apoptosis and inhibition of cell growth in A549 by regulating Ku70 acetylation (doi.org/10.3892/or.2016.4587).
Therefore, there are already some findings in literature that try to explain the mechanisms underlined the anticancer effect of EGCG. The authors should add this information in a more deeply way both in the introduction and discussion. Only in this context, they can outline the contribution of their work.

The quality of figures needs to be improved. More, in particular for the densitometric analyses of WB, change the colours because it was difficult for me to interpretate data.

Experimental design

Insert the company from each reagents/antibody/kit was purchased.
Insert a chapter of statistical analyses. All data do not have the significance. For example, ANOVA TEST, TT-TEST must be used to better evaluate the differences between samples.
Line 129. Indicate cells/well not cells/mL
The results need to be more discussed. For example, in the apoptosis section, indicate the percentage of cells on early and late apoptosis and the fold-increase with respect to untreated cells. More, compare the results with already obtained values in the other research (for example for A549 cells, doi.org/10.3892/or.2013.2933, doi.org/10.3892/or.2016.4587, doi: 10.3390/ijms222111833).

Validity of the findings

More literature data are needed to better understand and describe the author's contribution to this topic.

Additional comments

Please use the standard name of EGCG (Epigallocatechin-3-gallate and not epi-gallocatechin gallate as in the abstract).

---

## Round 0.2 · Minor Revisions

All major concerns have been addressed. Please address the additional set of comments provided the reviewers. I recommend minor revision to the manuscript.

·

Basic reporting

I commend the authors for their clear English used throughout the manuscript. In addition, the structure conforms to PeerJ standards. The introduction and background show context with referenced literature. Figures have now been labeled and carry detailed legends.
Raw data has been shared sufficiently. The correct provenance of the A549 and H1299 cell lines is described.

Experimental design

Original primary research within the scope of PeerJ. Research question is well defined and meaningful, and research fills an identified knowledge gap.
In addition, ethical standards are high and rigorous. Methods have now been described with sufficient details or information for reproducibility.

Validity of the findings

Thank you for providing the underlying data. Raw data for original OD value readings from the spectrophotometer have now been provided. Also, raw data after calculating %cell viability for each well using the OD values and formular have equally been provided. However, it is still not clear how the IC50 values were obtained. Were they extrapolated from graphs of calculated using a formular? Please provide clarifications on this.

Data provided has now been statistically analyzed in the study. To make the results more meaningful, the differences observed between the different groups have been determined with statistical significance using p-values (specifically, Figures 3 & 4). However, they are not reflected in the figures.

Figures 1 and 2 legends and the objective of the images (X10, X40, X100) have been included.
Figure 5 has equally been improved as the image legends provide more detail. Improvements have been done by describing the change in cell morphology observed as well as explaining what each quadrant represents.

Conclusions are well stated, linked to the original research question and limited to supporting results.

Additional comments

General comments
The paper investigates the anti-cancer potential of different doses of ECGC on the proliferation and inducing apoptosis in lung cancer cell lines A549 and H1299 in vitro. The authors claim that ECGC is safe, effective and exhibits significant potential in lung cancer therapy by reducing cell growth and promoting apoptosis with the PI3K/Akt signaling pathway. Thus, promising as a natural therapeutic agent in the treatment of lung cancer.

Abstract
Your abstract is now coherent with regards to the cell lines used in this study.
Keywords have now been provided.

Introduction
Too long.
Should be concisely written. A length of 3-4 paragraph maximum.
Please take out all irrelevant material. It looks to me like a thesis introduction whereas it is only a manuscript.

Materials and Methods
Cell lines, the reagents/kits as well the antibodies have now been mentioned and where they were purchased/obtained from. However, their catalogue numbers are still not provided.

Again, the MTT assay has been re-done using a 96 well plate as recommended, and all the raw data provided in the supplementary files. However, raw data reveals OD readings in 5 replicates, whereas methods section report data in triplicates. This is confusing. Can the authors clarify this?


Statistical Analysis
Data has now been statistically analyzed in a bid to effectively compare the differences observed in the different group treatments and to know if these differences are significant or not. However, this is not reflected in the figures. The analysis is not meant for the reviewer, neither is it meant to be kept in the supplementary files, it is meant to improve the interpretation and understanding of the presented results. Please can the authors insert the p-values and other necessary information in the graphs/figures? Besides, interpretation is still a major challenge here because looking at the p-values generated, the analysis does not demonstrate the difference observed between each group when compared to the control group. This makes it hard to understand the authors claim.

Reviewer 2 ·

Basic reporting

Following the reviewers' suggestions, the authors improve the scientific soundness of the manuscript. They also add technical details in the Materials and Methods section and statistical analyses, which are essential to demonstrate the validity of the results obtained.

Regarding literature references, the ref. Laudadio, Mangano and Minnelli should be inserted on line 142, at the end of the sentence "specifically targeting the epidermal growth factor receptor (EGFR), which is often overexpressed or mutated in cancer cells". Therefore, delete this citation from lines 133 and 145.

Experimental design

The authors inserted the company from each reagents/antibody/kit was purchased.

The authors have included a chapter on statistical analyses. However, they must add the statistical results in the figure (e.g. by using asterisks) and should also add the relative statistical significance in the figure captions (e.g. *P<0.001).

Validity of the findings

The authors have added more literature data and their contribution to EGCG as an anticancer polyphenol is now clearer.

Additional comments

English needs to be revised and improved. I suggest to the authors a correction by a native speaker.

---

## Round 0.3 · accepted · Accept

Thanks for addressing the reviewer's suggestions in this version.